# Determination and Measurement of Melanopic Equivalent Daylight (D65) Illuminance (*mEDI*) in the Context of Smart and Integrative Lighting

**DOI:** 10.3390/s23115000

**Published:** 2023-05-23

**Authors:** Vinh Quang Trinh, Peter Bodrogi, Tran Quoc Khanh

**Affiliations:** 1Laboratory of Adaptive Lighting Systems and Visual Processing, Technical University of Darmstadt, Hochschulstr. 4a, 64289 Darmstadt, Germany; khanh@lichttechnik.tu-darmstadt.de; 2ERCO GmbH, Brockhauser Weg 80-82, 58507 Lüdenscheid, Germany; p.bodrogi@erco.com

**Keywords:** non-visual effects of light, melanopic equivalent daylight (D65) illuminance, *mEDI*, melanopic equivalent daylight (D65) efficacy ratio, *mDER*, intelligent and integrated lighting, *mDER* calculation model, *mEDI* measurement, *mEDI* determination

## Abstract

In the context of intelligent and integrative lighting, in addition to the need for color quality and brightness, the non-visual effect is essential. This refers to the retinal ganglion cells (ipRGCs) and their function, which were first proposed in 1927. The melanopsin action spectrum has been published in *CIE S 026/E: 2018* with the corresponding melanopic equivalent daylight (D65) illuminance (mEDI), melanopic daylight (D65) efficacy ratio (mDER), and four other parameters. Due to the importance of mEDI and mDER, this work synthesizes a simple computational model of mDER as the main research objective, based on a database of 4214 practical spectral power distributions (*SPD*s) of daylight, conventional, *LED*, and mixed light sources. In addition to the high correlation coefficient $R^{2}$ of 0.96795 and the 97% confidence offset of 0.0067802, the feasibility of the mDER model in intelligent and integrated lighting applications has been extensively tested and validated. The uncertainty between the mEDI calculated directly from the spectra and that obtained by processing the *RGB* sensor and applying the mDER model reached ± 3.3% after matrix transformation and illuminance processing combined with the successful mDER calculation model. This result opens the potential for low-cost *RGB* sensors for applications in intelligent and integrative lighting systems to optimize and compensate for the non-visual effective parameter mEDI using daylight and artificial light in indoor spaces. The goal of the research on RGB sensors and the corresponding processing method are also presented and their feasibility is methodically demonstrated. A comprehensive investigation with a huge amount of color sensor sensitivities is necessary in a future work of other research.

## 1. Introduction

The development of lighting technology from the beginning of the 20th century to the present can be roughly divided into three stages, some of which overlap. Until the end of the 20th century, with thermal radiators and discharge lamps as light sources, the first component of integrative *HCL* (“*Human Centric Lighting*”) lighting technology was about “*visual performance*” (e.g., reaction time, contrast perception, reading speed, visual acuity) to enable visibility, improve work performance, and minimize error rates in industrial, educational, and healthcare facilities and offices, among others [1,2,3,4,5,6,7,8,9]. Since the 1970s and even more so since the mid-1990s, with the growing importance of education and the information society, the psychological effects of light (e.g., scene preference, room preference, user satisfaction, spatial perception, attractiveness, and color preference) have become more and more important. As a result of this research process, international and national standards (e.g., EN 12464 standard [9]) have been formulated for lighting design and the development of interior lighting systems. For special applications (e.g., exhibitions, meeting rooms in prestigious buildings, and treatment rooms in medical centers and hospitals) the following values in Table 1 [10,11,12,13,14,15,16,17] from research studies have been used.

In [18], De Boer and Fischer stated in 1978 that, for a horizontal illuminance of 1000 lx (which roughly corresponds to the preferred illuminance according to the above-mentioned study by Moosmann [13]), a luminance of 200 cd/m^2^ is preferred for the ceiling and 100 cd/m^2^ for the walls. At a horizontal illuminance of 500 lx, the above values can be estimated to be about 210 cd/m^2^ (ceiling) and 70 cd/m^2^ (walls). Another specification of the preferred luminance at 75 cd/m^2^ on the wall at a horizontal illuminance of $E_{v}$ = 500 lx was found in Balder’s 1957 publication [19]. Since the beginning of the 21st century, with the quantitative discovery of intrinsically photosensitive retinal ganglion cells (ipRGCs), other non-visual effects of light have been considered, such as circadian rhythm, hormone production and suppression, sleep quality, alertness, and mood. The overall picture of light effects on humans in the three contexts of “*visual performance*”, “*psychological light effects*”, and “*non-visual light effects*” is shown in Figure 1, which attempts to describe a chain of signal processing from the optical and temporal input parameters to the light users with influencing parameters to the output parameters expressing the physiological, behavioral, and human biological effects of light and lighting.

In recent years, there has been a large amount of research and publications on non-visual lighting effects that have been analyzed and reviewed in books and literature reviews [20,21,22,23]. In the majority of scientific publications cited in these reviews, the input parameters describing lighting conditions were photometric metrics (illuminance, luminance) and/or color temperature. In addition to the optimal lighting design of buildings and the development of modern luminaires according to the criteria of visual performance and psychological lighting effects, it is necessary to define suitable input parameters for the description of non-visual lighting effects to determine their optimal values and to measure these parameters in a wide variety of applications in the laboratory and in the field. To achieve the above goal, the following three research questions for lighting research must be formulated in general terms:Which input parameters can be used to describe the non-visual effects of light in their variety of manifestations (alertness, sleep quality, hormone production and suppression, and phase shift)?What values of these input variables are currently considered in the literature to be minimal, maximal, or optimal?Which measuring devices, sensor systems, and measuring methods can be used to measure the input quantities for non-visual lighting effects and process them in the context of smart lighting in the course of the control and regulation of *LED* or *OLED* luminaires on the basis of the definition of personal and room-specific applications?To define the parameters of non-visual lighting effects (Research Question 1), we must consider that, in the last 20 years, with more activities in the time range between 2016 and 2022, numerous international scientific discussions and analysis tasks have been carried out to specify the right metric for non-visual effects using datasets from experiments of different research groups on nocturnal melatonin suppression as a validation basis. Rea et al. defined and improved the CS model (Version 2018) in the two years 2020 and 2021, wherein they took into account the exposure time *t* (in hours) and the visual field, as well as modeled the contributions of the ipRGC channel, the *S* cones, the rods, and the (L+M) channel [24,25,26]. This improved formulation was validated using melatonin suppression datasets [26]. The measurement method for the CS parameter in the 2018 version was published by Truong et al. [27]. As well, the International Commission on Illumination (*CIE*) [28] and scientists in the fields of neurophysiology, sleep research, and lighting technology have recently made efforts to find effective parameters based on the evaluation of the five photoreceptor signals (LMS cones, rods, and ipRGC cells) and the corresponding calculation tool [29]. This scientific process resulted in the definitions of “*Melanopic Equivalent Daylight (D65) Illuminance (mEDI)*” and the “*Melanopic Daylight (D65) Efficacy Ratio (mDER)*”. These definitions have been recognized and used by international experts for several years and form the basis of Section 2 of this paper. Additionally, similar issues have been addressed in the WELL Building Standard v2, Q3 2020 version [30,31], which includes recommendations for melanopic equivalent daylight illuminance (mEDI≥136, 109, 218, or 180 depending on the type of light sources). To answer Research Question 2, we recognize that international sleep researchers and neuroscientists made some recommendations in 2022 regarding non-visual light effects, physiological aspects, sleep, and wakefulness based on their literature reviews [32]. The physiological aspects included hormonal regulation cycles, heart rate, core body temperature, and certain brain activities. The main findings were as follows:(a)“*Daytime light recommendations for indoor environments: Throughout the daytime, the recommended minimum melanopic EDI is 250 lux at the eye measured in the vertical plane at approximately 1.2 m height (i.e., the vertical illuminance at eye level when seated)*”.(b)“*Evening light recommendations for residential and other indoor environments: During the evening, starting at least 3 h before bedtime, the recommended maximum melanopic EDI is 10 lux measured at the eye in the vertical plane approximately 1.2 m height. To help achieve this, where possible, white light should have a spectrum depleted in short wavelengths close to the peak of the melanopic action spectrum*”.(c)“*Nighttime light recommendations for the sleep environment: The sleep environment should be as dark as possible. The recommended maximum ambient melanopic EDI is 1 lux measured at the eye*”.The first recommendation, with a minimum value of “*melanopic EDI = Melanopic Equivalent Daylight (D65) Illuminance, mEDI*” of 250 lx measured vertically at the observer’s eye, is relevant and of great interest for the professional sector during the day (offices, industrial halls, educational and health facilities). The other two recommendations are for the dark hours in the home. For the solution to Research Question 3 “*Determination and measurement of input parameters for non-visual lighting effects*”, which builds the focus of this present paper, two application areas can be targeted:The determination and measurement of non-visual parameters after the completion of new lighting installations and comparison with the specifications of the previous lighting design; or verification of the results of the development of new luminaires for *HCL* lighting in the lighting laboratory or in the field. For this purpose, absolute spectroradiometers for spectral radiance or spectral irradiance are used to calculate the parameters mEDI and mDER at different locations in the building. These two non-visual parameters cannot be measured directly with integral colorimeters and illuminance–luminance meters.The control or regulation of modern semiconductor-based lights (*LED*-*OLED*) and window systems (daylight systems) with the help of sensors in the room (e.g., presence sensors, position sensors, light and color sensors) [33]. In order to achieve a predefined value of non-visual parameters such as mEDI at a specific location in the room (e.g., in the center of the room or at locations further away from the windows), while taking into account dynamically changing weather conditions and the whereabouts of the room users, in practice, relatively inexpensive but sufficiently accurate optical sensors (RGB sensors) are required. The goal is to obtain not only the target photometric and colorimetric parameters such as illuminance $E_{v}$ (in lx), chromaticity coordinates (*x*, *y*, *z*), or correlated color temperature (CCT in K), but also the non-visual parameters mEDI and mDER. The principle of this Smart Lighting concept using RGB color sensors is illustrated in Figure 2. The methods for measuring non-visual parameters with low-cost but well-qualified RGB color sensors are the content of the Section 3 and Section 4 and the focus of this paper.The notations “*integrative lighting*” and “*human-centric-lighting*” used in this publication have been defined by the ISO/CIE publication [34] and mean a lighting concept and practice that integrates both visual and non-visual effects and produces physiological as well as psychological benefits for light users. The terms “*smart lighting*” or “*intelligent lighting*” are used in the same way and mean a framework that combines the above-mentioned integrative lighting as a goal and content of lighting practice with the technological aspects of lighting (signal communication, Internet of Things, sensor systems, control electronics, software that includes the methods of artificial intelligence, LED luminaires) while taking into account the individual needs of light users. In order to achieve the above objectives, the content of this paper is structured as follows:Section 2 defines the non-visual parameters according to the CIE publication [28].Section 3 describes the characterization and signal transformation mathematics for converting the RGB signals of the RGB sensors to the chromaticities *x*, *y*, and *z* according to the CIE publication [35] for a viewing angle of 2°.In Section 4, a computational model is established and analysed that converts the chromaticity *z* into the non-visual parameter mDER and the combination of *z* and illuminance $E_{v}$ into mEDI. Section 5 summarizes the results with some outlook.

## 2. Definition of Non-Visual Input Parameters [28]

The extensive mathematical treatment and definition of the non-visual input parameters are described in detail in the CIE publication *S 026/E:2018* “*CIE System for Metrology of Optical Radiation for ipRGC-Influenced Responses to Light*” [28], and those contents are summarized in this paper. The purpose of the CIE publication is to define spectral response curves, quantities, and metrics to describe optical radiation for each of the five photoreceptors in the human eye that may contribute to non-visual processes. The spectral sensitivity functions of the five receptor types are specified as follows (see Figure 3):S-cone-opic, $s_{10}\left(\lambda \right)$;M-cone-opic, $m_{10}\left(\lambda \right)$;L-cone-opic, $l_{10}\left(\lambda \right)$;Rhodopic, $V^{\prime }\left(\lambda \right)$, rods;Melanopic, mel(λ), ipRGCs.The CIE defined the α-opic quantities in Table 2 with the aid of the spectral sensitivities in Figure 3, and these quantities include the spectral radian flux $\Phi_{e,\lambda }\left(\lambda \right)$, the spectral radiance $L_{e,\lambda }\left(\lambda \right)$, and the spectral irradiance $E_{e,\lambda }\left(\lambda \right)$ (α represents one of the five photoreceptor types).

From this point, further parameters can be derived for assessing the non-visual effect of light, where $S_{\alpha }\left(\lambda \right)$ is the spectral sensitivity function of one of the five receptor types, $K_{m}$ is the known photometric radiation equivalent (=683.002 lm/W), and $\Phi_{D65,\lambda }\left(\lambda \right)$ is the spectrum of standard daylight type D65 (see the Equations (4) and (5)). The quantity “$K_{\alpha ,v}$” is the α-opic efficacy of the luminous radiation.
(4)$$K_{\alpha ,v}=\frac{\int \Phi_{e,\alpha }\left(\lambda \right)\;\cdot \;S_{\alpha }\left(\lambda \right)\;\cdot \;d\lambda }{K_{m}\;\cdot \;\int \Phi_{e,\alpha }\left(\lambda \right)\;\cdot \;V_{\alpha }\left(\lambda \right)\;\cdot \;d\lambda }$$Equation (4) is similar to Equation (3.4) on Page 4 of [28].
(5)$$K_{\alpha ,v}^{D65}=\frac{\int \Phi_{D65,\alpha }\left(\lambda \right)\;\cdot \;S_{\alpha }\left(\lambda \right)\;\cdot \;d\lambda }{K_{m}\;\cdot \;\int \Phi_{D65,\alpha }\left(\lambda \right)\;\cdot \;V_{\alpha }\left(\lambda \right)\;\cdot \;d\lambda }$$Equation (5) is similar to Equation (3.7) on Page 5 of [28]. For melanopic efficacy (α = mel), the parameter value $K_{mel,V}^{D65}$ equals 1.3262 mW/lm. Consequently, the parameters mEDI and mDER can be set up as shown in Table 3. If one wants to interpret the meaning of mEDI or mDER for lighting engineering, there are two main aspects:mEDI is the illuminance of the standard daylight illuminant D65 that has as much melanopic efficacy as the test light source for a given illuminance $E_{v}$ (lx) caused by the test light source. See Equation (6) in Table 3.mDER is the ratio of the illuminance of the standard illuminant D65 (mEDI) to the illuminance of the test illuminant $E_{v}$ (in lx) when the absolute melanopic efficacy of both illuminants is set equal. See Equation (9) in Table 3.The melanopically effective parameters mDER and mEDI can be calculated from the Equations (3), (5), (6), and (9) if the spectral irradiance $E_{e,\lambda }\left(\lambda \right)$ is known from a spectroradiometric measurement. The key question in this publication is whether these two parameters can also be determined with sufficient accuracy in practical lighting technology using a well-characterized and inexpensive RGB sensor in the sense of intelligent lighting technology (smart lighting). Section 3 deals with the description of the RGB sensor and how the RGB sensor signals can be transformed into tristimulus values (RGB) and into chromaticity coordinates (x,y,z) by means of a comprehensive spectral analysis.

## 3. *RGB* Color Sensors: Characterization and Signal Transformation

After the definition of the non-visual input parameters [28] in Section 2, we can understand the main concepts and mathematical forms of mEDI and mEDR. In Section 4, the prediction of a simple computational model for the non-visual quantities mEDI and mDER was carried out. As well, the verification of the feasibility of a RGB sensor using this model was implemented to check the synthesized prediction model for lighting applications. This was also performed to confirm that the model not only conformed to the mathematical calculations, but also to determine whether it can be applied in the lighting systems with low-cost RGB color sensors. Therefore, Section 3 must attempt to describe RGB color sensors, their characterization, and appropriate signal transformation techniques. This section serves as a link between Section 2 and Section 4, as well as to balance the essential material for the verification of the prediction model using a color sensor as an example in Section 4 later. As a demonstration of the methodology for processing RGB sensors, it is not necessary to collect all color sensor sensitivities, synthesize, and compare them, but the most important thing here is to prove that the methodology can work well in this approach. Consequently, the future work can be implemented more comprehensively for different color sensor families through other research.

### 3.1. Characterisation of RGB Color Sensors

The color sensors (see Figure 4) include arrays of individual sensors or a group of sensors based on semiconductors with silicon as the semiconductor material, which are most commonly used for the visible spectral range between 380 nm and 780 nm.

In order to generate the corresponding spectral sensitivity curves of the individual color channels in the red, green, and blue range (so-called RGB channels), optical color filters based on thin-film technology (interference filters), absorption glass filters or color varnishes were microstructured and applied to the respective silicon sensor. For the correct spatial evaluation of the optical radiation according to the cosine law, a so-called cos prefix (usually a small diffuser plate made of optical scattering materials) was applied to the RGB sensor. The photons of each wavelength are absorbed by the RGB sensor, and photon currents are generated, which are then converted into voltages in an amplifier circuit. These voltages are then digitized in different bit depths (8 bits, 12 bits, 16 bits) by an A/D converter (analog–digital). The spectral sensitivity of each color channel *R*–*G*–*B* is, therefore, made up of the components shown in the Equations (10)–(12).
(10)$$R\left(\lambda \right)=S\left(silicon,\lambda \right)\;\cdot \;\tau \left(diffusor,\lambda \right)\;\cdot \;\tau \left(\mathrm{color}\;{\mathrm{lacquer}}_{R},\lambda \right)\;\cdot \;K_{R}=\frac{CV_{R}\left(\lambda \right)}{E(\lambda )}$$
(11)$$G\left(\lambda \right)=S\left(silicon,\lambda \right)\;\cdot \;\tau \left(diffusor,\lambda \right)\;\cdot \;\tau \left(\mathrm{color}\;{\mathrm{lacquer}}_{G},\lambda \right)\;\cdot \;K_{G}=\frac{CV_{G}\left(\lambda \right)}{E(\lambda )}$$
(12)$$B\left(\lambda \right)=S\left(silicon,\lambda \right)\;\cdot \;\tau \left(diffusor,\lambda \right)\;\cdot \;\tau \left(\mathrm{color}\;{\mathrm{lacquer}}_{B},\lambda \right)\;\cdot \;K_{B}=\frac{CV_{B}\left(\lambda \right)}{E(\lambda )}$$
where:S(silicon,λ) is the spectral sensitivity of the silicon sensor;τ(diffusor,λ) is the spectral transmittance of the cos diffusor;$\tau (\mathrm{color}\;{\mathrm{lacquer}}_{i},\lambda )$ is the spectral transmittance of the color filter layer for the respective color channel *i* = *R*, *G*, *B*;$K_{R}$, $K_{G}$, and $K_{B}$ are absolute factors to account for current-to-voltage conversion and voltage digitization;$CV_{i}\left(\lambda \right)$ values are output signals (analog or digital) of the respective *R*–*G*–*B* color sensor at wavelength λ.If a certain spectral irradiance E(λ) is present on the RGB sensor, three corresponding output signals R, G, and B are generated in the three color channels *R*, *G*, and *B*, respectively (see the Equations (13)–(15)).
(13)R=∫E(λ)·R(λ)·dλ
(14)G=∫E(λ)·G(λ)·dλ
(15)B=∫E(λ)·B(λ)·dλFor the calculation of the RGB channel signals, it is, therefore, necessary to know or to determine the spectral sensitivity function of each individual color channel *R*, *G*, *B* by laboratory measurements. Consequently, the spectral apparatus for determining the spectral sensitivity of the semiconductor sensors in the authors’ light laboratory (see Figure 5) consists of a high-intensity xenon ultrahigh-pressure lamp, a grating monochromator (spectral half-width Δλ = 2 nm, spectral measuring steps for the entire spectrum between 380 nm and 780 nm, Δλ = 2 nm), and an integrating sphere for homogenizing the quasi-monochromatic radiation coming out of the monochromator. The RGB color sensor and a known calibrated reference sensor are located at two different locations on the inner wall of the sphere behind a shutter.

During spectral measurement of the RGB sensor, one can determine the spectral sensitivity for the red color channel, for example, according to Equation (16).
(16)$$R\left(\lambda \right)=\frac{CV_{R}\left(\lambda \right)}{E(Sphere,\lambda )}$$
with $CV_{R}\left(\lambda \right)$ as the output signal of the red color channel and E(sphere,λ) as the spectral irradiance at the inner wall of the sphere when the monochromator is set to the wavelength λ, which in turn can be determined from the measured output photocurrent i(reference,λ) and the known spectral sensitivity of the reference sensor S(reference,λ). As a result, we can write Equation (17).
(17)$$E\left(\mathrm{sphere},\lambda \right)=\frac{i(\mathrm{reference},\lambda )}{S(\mathrm{reference},\lambda )}$$The absolute spectral sensitivity of each color channel *R*, *G*, *B* can be determined from the Equations (16) and (17). Figure 6 and Figure 7 show examples of the spectral response curves of some RGB color sensors measured in the authors’ light laboratory. In Figure 6, the two datasets of the same type, SeS1 and SeS2, differ by the different peak heights of the *R* channel and by the different slopes of the *G* and *B* curves, because the two sensors SeS1 and SeS2 came from different production sets. The SeS1 and SeS2 spectral sensitivity curves in Figure 6 are fundamentally different from the RGB spectral sensitivity curves of the color sensor type in Figure 7.

The most accurate characterization possible for RGB sensors and the best possible similarity between CIE color matching functions CIExyz(λ) for a field of view of 2° and RGB(λ) sensitivities are very important and have a direct influence on the later determination of non-visual parameters such as the results in Table 13. The smaller the difference between RGB(λ) sensitivities and CIE color matching functions CIExyz(λ) for a field of view of 2°, the closer the distance between the values $RGB_{SeS.}$, the chromaticity coordinates $xyz_{CIE}$, and later the values $xyz_{SeS.}$ after matrix transformation of the chromaticity coordinates $xyz_{CIE}$. Finally, the difference between the determined $z_{SeS.}=\left(1-x_{SeS.}-y_{SeS.}\right)$ and the CIE chromaticity coordinates $z_{CIE.}=\left(1-x_{CIE.}-y_{CIE.}\right)$ must be as small as possible, and this is understood as the primary uncertainty of the non-visual parameter calculation. Then the non-visual parameter model $mEDR=f(z_{SeS.})$ in Section 4 and the parameter $mEDI=[mEDR\left(z_{SeS.}\right)\;\cdot \;E_{v}]$ determine the secondary uncertainty of the non-visual parameter calculation. The abbreviation “*SeS.*” indicates that it is the value of the sensors.

### 3.2. Method of Signal Transformation from RGB to XYZ


Figure 6 and Figure 7 show that the RGB curves of the real color sensors differ more or less strongly from the xyz curves of the CIE color matching function for a field of view of 2° [35]. In order to obtain the XYZ tristimulus values, the generated RGB signals must be transformed into digital form by matrixing (Equation (18)).
(18)$$\left(\begin{array}{c} X_{\mathrm{SeS}.,i}\\ Y_{\mathrm{SeS}.,i}\\ Z_{\mathrm{SeS}.,i} \end{array}\right)=\left(\begin{array}{cccc} m_{1,1} & m_{1,2} & \ldots & m_{1,n}\\ m_{2,1} & m_{2,2} & \ldots & m_{2,n}\\ m_{3,1} & m_{3,2} & \ldots & m_{3,n} \end{array}\right)\;\cdot \;\left(\begin{array}{c} R_{1,i}\\ G_{1,i}\\ B_{1,i}\\ \ldots \end{array}\right).$$
the sensor technology, the amplification electronics, and the A/D conversion often have a more or less pronounced non-linear behavior, it may be necessary to use different matrixing types from 3 × 3 to 3 × 22. See Table 4.

If the sensor electronics are linear and the spectral response curves have a similar relative shape to the xyz color matching functions or LMS sensitivity spectra of the retinal photoreceptors, the 3 × 3 to 3 × 8 matrices will yield contributions from RGB signals as a function of the first term. The more the sensor electronics deviate from linear behavior, the more the shape of the spectral curves of the real sensors deviate from the xyz color matching functions. Therefore, matrices with RGB contributions in quadratic or cubic functions (see Table 4) should be set up. The procedure for finding the optimal matrix based on the chromaticity difference $\Delta u^{\prime }v^{\prime }$ is shown in Figure 8.

The goal of the optimization is to get the chromaticity (*x*, *y*, *z*) of the RGB sensor as close as possible to that derived from the CIE calculation. Instead of working with the non-uniform xy color diagram, it is better to work with the more uniform $u^{\prime }v^{\prime }$ diagram (*CIE*, 1964). Bieske [37] showed in her dissertation that, when $\Delta u^{\prime }v^{\prime }$ is less than $10^{-3}$, the color difference is not perceived by the subjects. From $10^{-3}$ to $3\times 10^{-3}$, the color difference can be perceived but is still acceptable. If this value is higher than $5\times 10^{-3}$, the color difference is unacceptable to the subjects. Based on this scientific contribution, the optimization with $u^{\prime }v^{\prime }$ not only gives the best *z*-value, but can also be directly checked and compared with the obtained perception thresholds.

### 3.3. Matrix Transformation in Practice and Verification with a Real RGB Color Sensor

For the color sensor type whose spectral sensitivity curves are shown in Figure 7, the optimization processes are carried out according to the scheme in Figure 8 with a series of matrix types from 3 × 3 to 3 × 22 with nine different lamp spectra (as a training set). These nine lamp spectra are tabulated in Table 5 and shown in Figure 9. The illuminance for the optimization of the matrix was chosen to be $E_{v}$ = 750 lx. In this training set, the following lamp types were selected to reflect the variety of types used in practice today: fluorescent lamps between 2640 K and 4423 K with many spectral lines (see Figure 9), a halogen incandescent lamp, two phosphor-converted white LEDs, and a combination of RGBLEDs and warm white LEDs (RGBWW4500K).

The calculation with different matrix types and with individual lamp spectra yields color differences $\Delta u^{\prime }v^{\prime }$ of varying magnitudes, with the 3 × 3 matrix form yielding the smallest color difference (see Table 6). Optimization with more lamp types did not bring any significant improvement in this case. The final 3 × 3 optimized matrix for the transformation from RGB to XYZ, as well as the formula for the calculation of the illuminance from the RGB signals, are shown in Table 7.

To verify the prediction quality of the formula in Table 7, eight spectra were selected in the next step. These were white phosphor converted LEDs (*pc-LEDs*) and a mixture of daylight spectra with white phosphor converted LEDs (*TL-pc-LEDs*), as is often the case in practical indoor lighting (see Figure 10). The illuminance was again set to 750 lx. The results of the check are shown in Table 8. From Table 8 it can be seen that
(a)The deviation of the illuminances, calculated directly from the lamp spectra via the RGB color sensor signals and via the formula in Table 7, was below 0.65% in percentage terms;(b)For the majority of *pc-LEDs* and for the combinations daylight–white LEDs, the chromaticity difference $\Delta u^{\prime }v^{\prime }$ was in a small or moderately small range from the point of view of practical lighting technology, although the matrix transformation was synthesized only for the general case of many lamp types. No specific procedure was optimized for the specific case of only mixed light between daylight and conventional light, but the color and illuminance difference was still very small when using the achieved matrix transformation. Therefore, the authors did not build a separate case of mixed light with specific matrix transformation. Also, *pc-LEDs* and their mixing with daylight are dominant nowadays, so this verification in this way makes sense for applications. Therefore, the extreme cases of very complicated spectral shapes are not necessary to investigate their feasibility, because they are very rarely applied in today’s life. And an exception is the spectrum R12GB12-5000 K (green curve in Figure 10) with three distinct peaks of the three RGBLEDs (*B* around 455 nm, *G* around 525 nm, *R* around 660 nm), which is a very good demonstration example of a complicated spectral form. In this case, the color difference was $\Delta u^{\prime }v^{\prime }=10.1\times 10^{-3}$.
However, for the case of pure daylight spectra, a special matrix was found and described in Table 9 as a specific case. For the investigated RGB color sensor type, 185 daylight spectra were measured absolutely on a sunny summer day (19 August 2020 in Darmstadt, Central Europe) from 6:32:00 in the early morning to 20:35:32 in the late evening (see Figure 11). The chromaticity and illuminance values were calculated and used as a training set for the matrix. The largest color difference with a 3 × 3 matrix was found to be $\Delta u^{\prime }v^{\prime }=1.2\times 10^{-3}$, which was recorded in the last minutes of the evening before sunset when the correlated color temperature was very high, measuring approximately in the range of 17,000 K on the day of measurement. Most of the color differences were well below this value.

To verify the 3 × 3 matrix for daylight, 24 daylight spectra measured on a overcast day (23 September 2020 in Darmstadt, Germany) were considered. The verification results are shown in Table 10 with 9 out of 24 spectra as examples. There, the chromaticities and illuminances determined from the measured spectra are compared with the data obtained by matrixing the RGB color sensor (processed data). The calculated color difference $\Delta u^{\prime }v^{\prime }$ was very small and lay in the range of $10^{-4}$.

The determination of the optimal matrix for the transformation of RGB sensor signals into XYZ values and the illuminance $E_{v}$ described so far, as well as the verification results with an actual RGB color sensor type, prove that it is possible to obtain the colorimetric and photometric values XYZ, $E_{v}$, and CCT with relatively good results in the sense of a reliable, relatively accurate and adaptive lighting technology using inexpensive and commercially available color sensors. The next section deals with the accurate determination of the non-visual input quantities mEDI and mDER indoors (daylight and artificial light combined or separated) and outdoors during the day (with daylight only) from the tristimulus values XYZ and the illuminance $E_{v}$ of a qualified RGB color sensor, because, in practical lighting technology, a spectroradiometer is often too expensive and too cumbersome to handle.

## 4. Prediction of a Simple Computational Model for the Non-Visual Quantity mDER and Verification of the Feasibility of a RGB Sensor Using this Model

Once the matrices for electric light sources and for daylight spectra have been found, the tristimulus values XYZ, the chromaticity coordinates xyz, the color temperature CCT, and the illuminance $E_{v}$ can be obtained from the RGB color sensors used (see the Table 7, Table 8, Table 9 and Table 10). To obtain the melanopic equivalent daylight illuminance mEDI, one only needs to know the value of mDER and the illuminance $E_{v}$ (in lx) according to Equation (7), where the illuminance can be formed from the RGB signals (see the formula in Table 7 and Table 9) for determination. Consequently, the most important task is to calculate the dimensionless input quantity mDER (see Equation (9)) from the chromaticity coordinates *x*, *y*, *z* (*z* turned out to be the most predictive, so it was the coordinate used).

For this transformation from *z* to mDER, a large number of measured and simulated light source spectra were analyzed (see Table 11). The first four light source groups were real measured light sources from thermal radiators (28 light sources), compact and linear fluorescent lamps (252 light sources), different LED configurations (419 light sources), and 185 measured daylight spectra. These 884 light source spectra are shown in Figure 12. In order to simulate the combination of daylight with white LEDs or daylight with white fluorescent lamps, which can correspond to the lighting conditions of interiors with white LEDs or white fluorescent lamps with windows over a longer period of use, the authors of this paper took 185 measured daylight spectra and mixed them with a market-typical white LED spectrum or with a typical fluorescent lamp spectrum in nine different mixing ratios from 10% to 90%. This can be seen in Figure 13 for the case of combining daylight spectra with white LEDs. For the fluorescent lamps, it was similar, so a graphical representation of the mixing spectra is omitted here. The 185 daylight spectra with nine mixing ratios each resulted in 1665 simulated spectra each for the LED and fluorescent lamps (185 × 9 = 1665).

Using the 4214 light source spectra summarized in Table 11, 4214 mDER values and 4214 *z* values were calculated, from which their relation was determined. The formula for mDER is given in Equation (25).
(25)$$mDER=a\;\cdot \;e^{b\;\cdot \;z}-c\;\cdot \;e^{d\;\cdot \;z}$$The parameters *a*, *b*, *c*, and *d* for the fitting function and the correlation coefficients $R^{2}$ are given in Table 12. The course of the correlation between mDER (ordinate) and the chromaticity coordinate *z* (abscissa) is shown in Figure 14, where the fitting function follows the green course of the curve.

To check the predictive quality of Equation (25) and the transformation from *z* to mDER, the 4214 spectra in Table 11 were first used to calculate the 4214 mDER values from the spectra themselves (denoted as $mEDR_{\left[CIES026/E:2018\right]}$) and then another 4214 mDER values indirectly via Equation (25), with *z* as the independent variable (denoted as mEDR=f(z)) (see Figure 15). The goodness of fit ($R^{2}$) was 0.97844, and the linear constants were 1.0214 and −0.05712. Compared to the ideal constants of 1 and 0, the quality of the fit is very good.

To verify the quality of the characterization of the RGB color sensor under test (see Figure 7), the quality of the “*matrixing*” in the transformation from RGB to XYZ and $E_{v}$ (see Table 7) and the quality of the transformation from *z* to mDER (see Equation (25)) and finally the calculation of mEDI (see the Equations (6) and (9)), we set each of the eight light source spectra in Figure 10 to 750 lx. From these 8 spectra, the chromaticity coordinates *x* and *y* and the quantities CCT, $CRI\;R_{a}$, $E_{v}$, $mDER_{original}$, and $mEDI_{original}$ were calculated. They are listed in Table 13. For these eight spectra, the chromaticity coordinates $x_{new}$ and $y_{new}$, the value of $E_{v,new}$, and the values $mDER_{new}$ and $mEDI_{new}$ were also calculated from the RGB color sensor values by matrixing and transforming *z* to mDER and mEDI, respectively. The maximum relative deviations ΔmDER (in %) and ΔmEDI (in %) were found to be in the range of ±3.3%.

## 5. Conclusions and Discussion

In indoor lighting technology, including daylight components during the day, the daily task is to evaluate the lighting systems after the completion of the building or after reconstruction to assess the photometric and colorimetric quality of the newly developed interior luminaires and, in the context of intelligent lighting (smart lighting, HCL lighting, integrative lighting), to adaptively control the lighting systems in order to provide the room users with the best visual and non-visual conditions, as well as room atmosphere, at all times. In addition to the criteria of visual performance according to current national and international standards (e.g., EN DIN 12464 [9] or WELL [31]), aspects of psychological and emotional lighting effects, as well as the evaluation and adaptive control of lighting systems according to non-visual quality characteristics, should be considered. For this purpose, non-visual input parameters such as mEDI and mDER need to be measured and processed with sufficient accuracy, reliability, and reasonable effort. Existing illuminance meters, luminance meters, color and luminance cameras, and small portable colorimeters are currently only capable of measuring photometric quantities such as illuminance $E_{v}$, luminance $L_{v}$, colorimetric parameters such as chromaticity coordinates xyz, and correlated color temperature (CCT). To measure the non-visual parameters, a transformation is needed to convert the tristimulus values to the non-visual parameters, and a sensor platform is needed to convert the RGB sensor signals to the tristimulus values XYZ and $E_{v}$ (in lx). The authors of this paper have characterized some exemplary RGB color sensors on the basis of laboratory measurements, calculations, and optimizations by creating one or more matrices for the optimal transformation of RGB sensor signals to the tristimulus values. For different spectral sensitivities of the RGB color sensors, different matrices with linear or higher order RGB-signals can be optimized. As an example, for a certain RGB sensor, a 3 × 3-matrix was found by means of optimization with nine different spectra to be the best transformation that delivered a maximal chromaticity difference of Δu’v’=0.0083 (see Table 6). This matrix was verified with eight white phosphor-converted LEDs (*pc-LEDs*) and mixture of daylight spectra with white phosphor-converted LEDs (*TL-pcLEDs*) (see Table 8). For the spectra of daylight only, a special matrix was found that allowed for a much lower chromaticity difference (maximal $\Delta u^{\prime }v^{\prime }=0.000857$, see Table 10). The reconstruction of the illuminance (in lux) from the RGB signals delivered a difference in the order of 0.64% (see Table 8). As next steps, on the basis of extensive calculations, a transformation from the chromaticity coordinate *z* to mDER and, via $E_{v}$, also to mEDI, was achieved with good accuracy. Table 13 has shown the verification results with eight light source spectra, which were in the range of ΔmEDI=±3.3%, which can be accepted from a practical point of view of lighting engineers, lighting designers, and luminaire system developers. This work and the methodology described therein enable an accurate and financially justifiable assessment of lighting installations according to non-visual criteria, which will become increasingly important in the coming decades. This paper followed the aim to deliver, firstly, a framework and a general method on how to use commercially available RGB sensors to qualify them for measuring the metrics for non-visual effects. The results had some limitations, because only one RGB sensor type has been used in the methodical frame of this paper. Other RGB sensor types may have other spectral, optical, and electronic characteristics with different dark currents, signal-to-noise ratios, and cross-talk effects, which should require other forms of matrices (e.g., matrices with higher orders of RGB signals). Further studies will be planned by the authors of this present paper, in which the accuracies of several commercial RGB sensor types will be compared to the measurement results of an absolute measuring spectroradiometer for different test conditions (e.g., outdoor and indoor lighting conditions with different mixture ratios of daylight and artificial light sources).

## Figures and Tables

**Figure 1 sensors-23-05000-f001:**
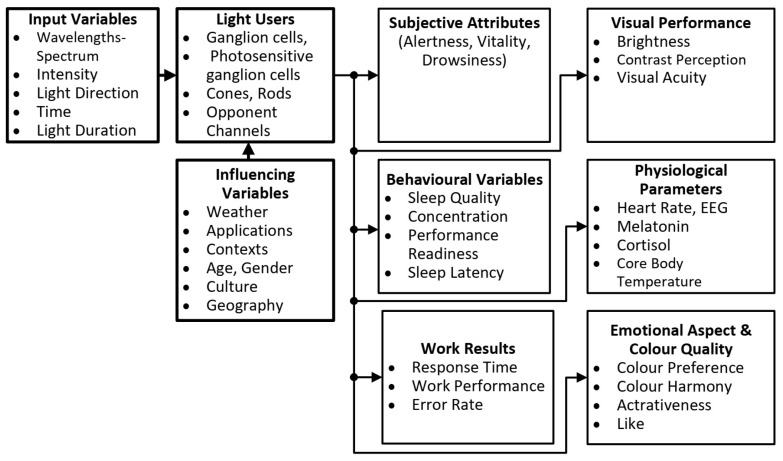
Input variables, influencing factors, and output variables in a comprehensive view of the effects of light on humans.

**Figure 2 sensors-23-05000-f002:**
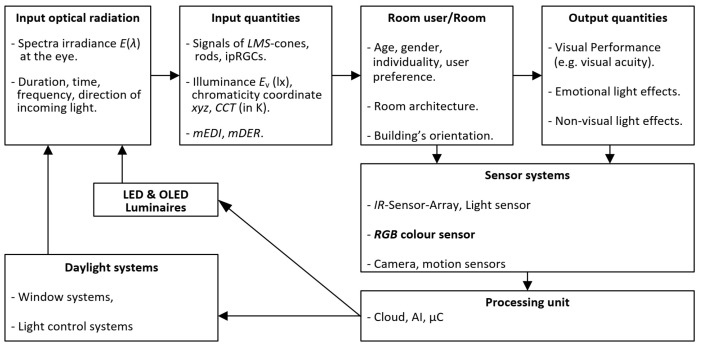
Principle of a Smart Lighting concept with RGB color sensors.

**Figure 3 sensors-23-05000-f003:**
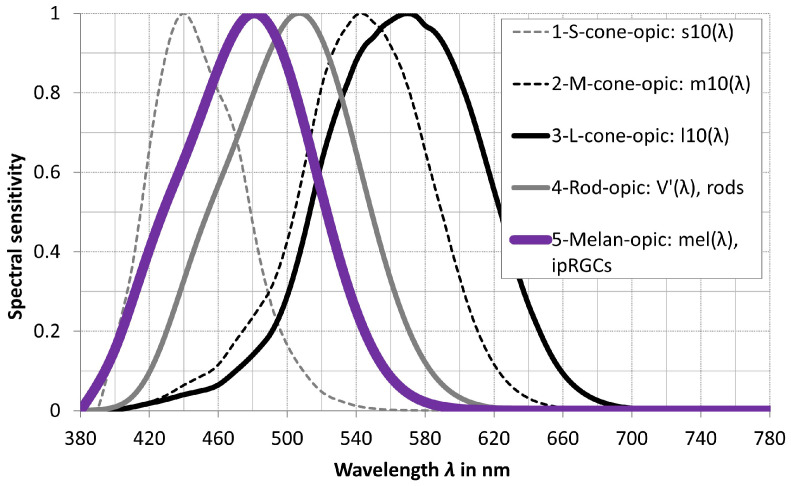
Five relative photoreceptor sensitivities in *CIE S 026/E:2018* [28].

**Figure 4 sensors-23-05000-f004:**
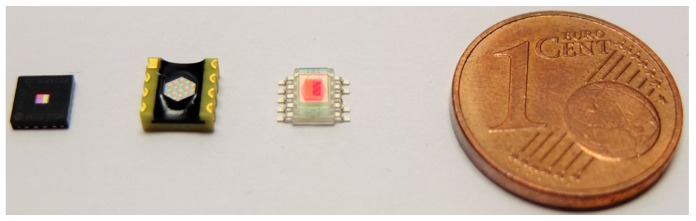
4 Examples of RGB color sensors. From left to right, the color sensor chips MTCS-CDCAF, MRGBiCS, and S11059 are shown. (Image Source: TU Darmstadt).

**Figure 5 sensors-23-05000-f005:**
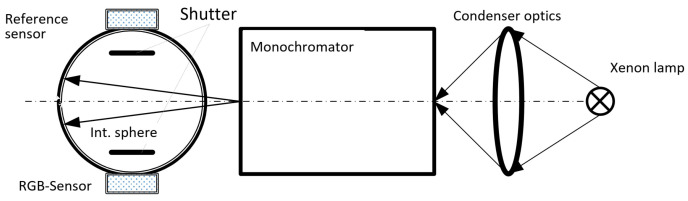
Schematic arrangement for measuring the spectral sensitivity of RGB sensors.

**Figure 6 sensors-23-05000-f006:**
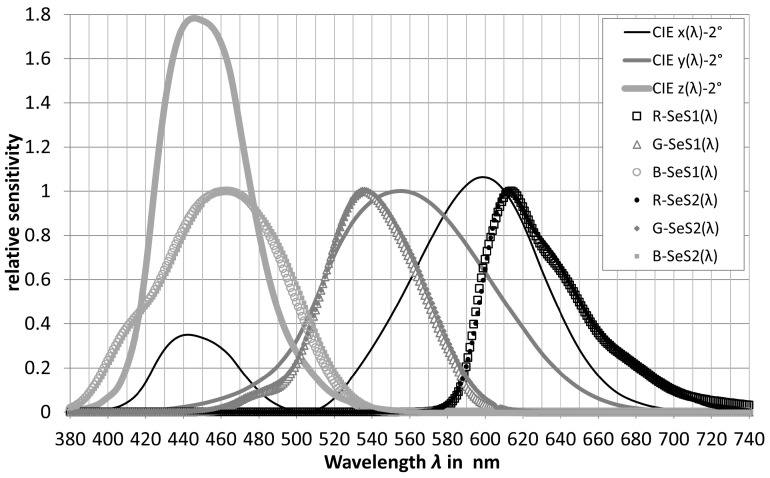
Examples of the spectral sensitivity curves of some RGB color sensors.

**Figure 7 sensors-23-05000-f007:**
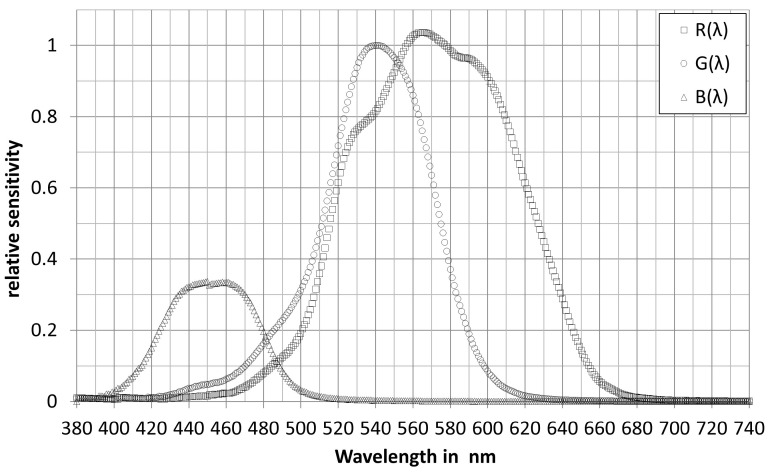
Spectral sensitivity curves of another RGB color sensor type.

**Figure 8 sensors-23-05000-f008:**
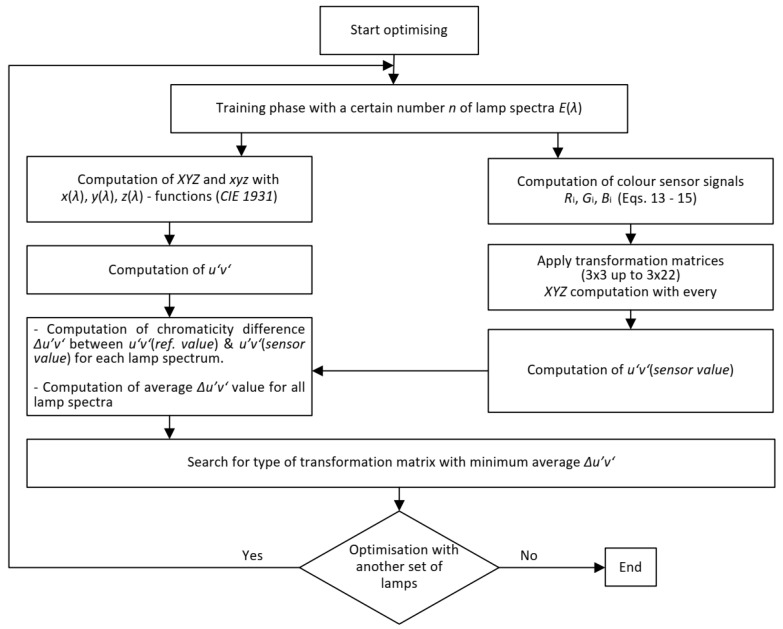
Flowchart for the optimization of a transformation matrix from RGB to XYZ based on the chromaticity difference $\Delta u^{\prime }v^{\prime }$ [35].

**Figure 9 sensors-23-05000-f009:**
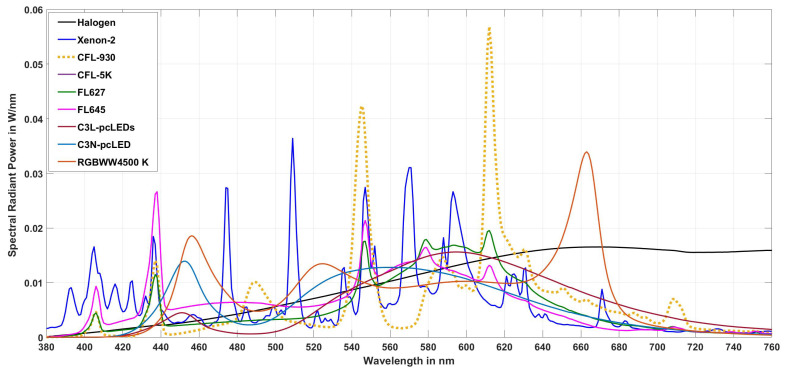
Spectra of the lamp types for matrix optimization.

**Figure 10 sensors-23-05000-f010:**
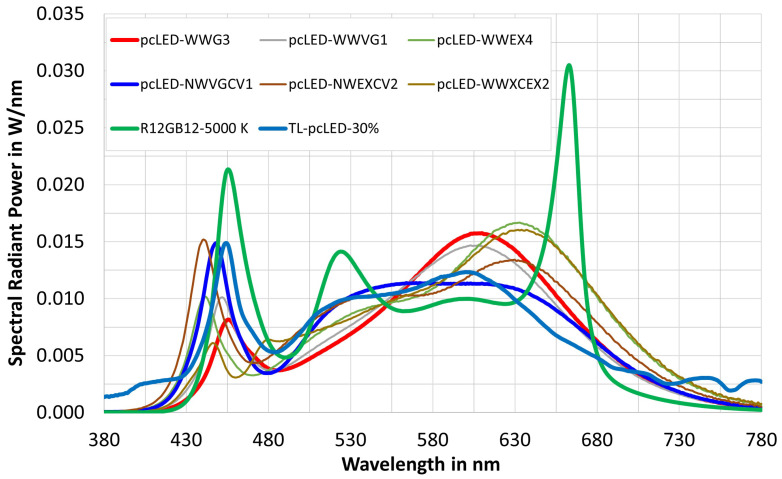
Lamp spectra of 8 light source types for the validation of the 3 × 3 matrix in Table 7.

**Figure 11 sensors-23-05000-f011:**
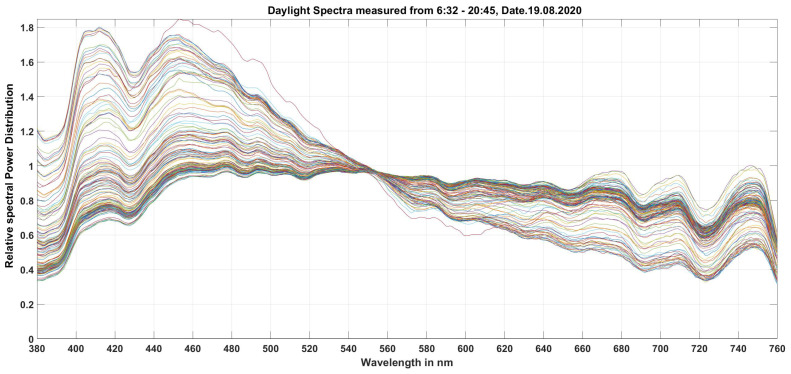
Some daylight spectra on a sunny day in Darmstadt, Germany (on 19 August 2020).

**Figure 12 sensors-23-05000-f012:**
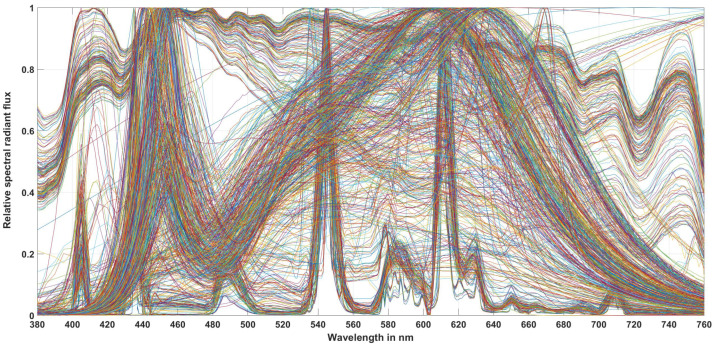
Spectra of 884 measured light source spectra.

**Figure 13 sensors-23-05000-f013:**
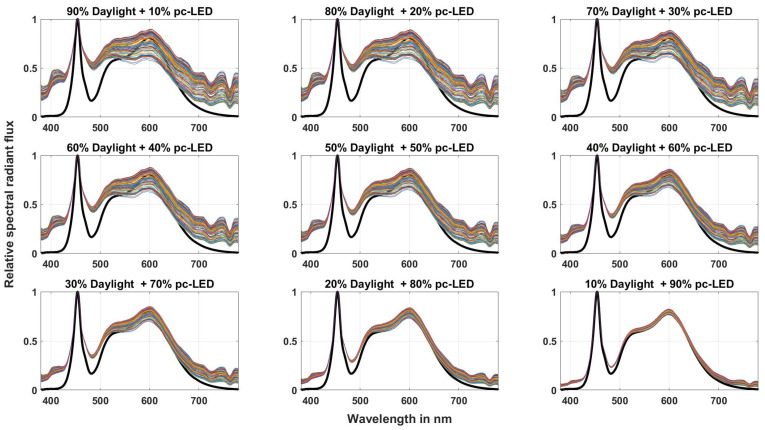
Spectra of 185 phases of daylight, mixed with white LEDs in 9 mixture ratios from 10% until 90%.

**Figure 14 sensors-23-05000-f014:**
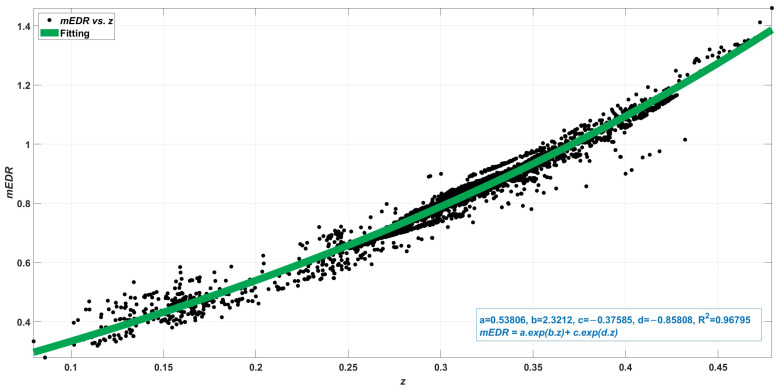
Correlation between mDER (ordinate) and the chromaticity coordinate *z* (abscissa).

**Figure 15 sensors-23-05000-f015:**
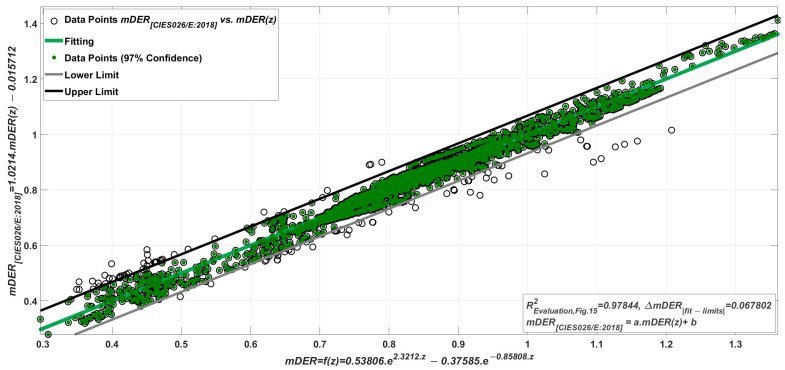
Comparison of the 4214 mDER values on the ordinate, calculated from the spectra themselves (denoted as $mEDR_{\left[CIES026/E:2018\right]}$), with the 4214 mDER(=f(z)) values calculated using Equation (25). $R^{2}=0.97844$. The linear constants a=1.0214, b=−0.015712 compared to the ideal constants 1 and 0 and the ΔmEDR 97% confidence interval of 0.067802 show a good quality.

**Table 1 sensors-23-05000-t001:** Parameters for interior lighting in [10,11,12,13,14,15,16,17].

No.	Parameter	Preferred Values
1	Horizontal illuminance $E_{v,h}$ in lux [11,12,13]	Greater than 850 lx; Recommended range: 1300 lx–1500 lx
2	Correlated color temperature CCT in K [14,15]	4000 K–5000 K
3	Color rendering index CIECRI [17]	>87
4	Indirect to total illuminance ratio γ [10,16]	>0.6–0.8

**Table 2 sensors-23-05000-t002:** α-opic quantities in *CIE S 026/E:2018* [28].

α-Opic Quantities		
**Parameter**	**Equation**	**Equation No.**
α-opic-radian flux	(1) $$\Phi_{e,\alpha }\;\mathrm{or}\;\Phi_{\alpha }=\int \Phi_{e,\alpha }\left(\lambda \right)\;\cdot \;S_{\alpha }\left(\lambda \right)\;\cdot \;d\lambda$$	similar to (3.1) Page 4 [28]
α-opic-radiance	(2) $$L_{e,\alpha }\;\mathrm{or}\;L_{\alpha }=\int L_{e,\alpha }\left(\lambda \right)\;\cdot \;S_{\alpha }\left(\lambda \right)\;\cdot \;d\lambda$$	similar to (3.5) Page 4 [28]
α-opic-irradiance	(3) $$E_{e,\alpha }\;\mathrm{or}\;E_{\alpha }=\int E_{e,\alpha }\left(\lambda \right)\;\cdot \;S_{\alpha }\left(\lambda \right)\;\cdot \;d\lambda$$	similar to (3.6) Page 5 [28]

**Table 3 sensors-23-05000-t003:** Melanopic equivalent D65 quantities of *CIE S 026/E:2018* [28].

α-Opic Quantities		
**Parameter**	**Equation**	**Equation No.**
Melanopic Equivalent Daylight (D65) Illuminance (*mEDI*) in lx	(6) $$mEDI=\frac{\int_{380}^{780}S_{mel.}\left(\lambda \right)\;\cdot \;E_{e,\lambda }\left(\lambda \right)\;\cdot \;d\lambda }{K_{mel.,V}^{D65}}$$	similar to (3.9) Page 6 [28] with α=mel.
	(7) $$mDER=\frac{K_{mel.,V}}{K_{mel.,V}^{D65}}\left(*\right)$$	↓
Melanopic Daylight (D65) Efficacy Ratio (mDER)	(8) $$mDER=\frac{1}{K_{mel.,V}^{D65}}\;\cdot \;\frac{\int_{380}^{780}S_{mel.}\left(\lambda \right)\;\cdot \;E_{e,\lambda }\left(\lambda \right)\;\cdot \;d\lambda }{E_{v}}$$	↓
	(9) $$mDER=\frac{\frac{1}{K_{mel.,V}^{D65}}\;\cdot \;\int_{380}^{780}S_{mel.}\left(\lambda \right)\;\cdot \;E_{e,\lambda }\left(\lambda \right)\;\cdot \;d\lambda }{E_{v}}$$	similar to (3.10) Page 7 [28] with α=mel.
(*) Note: For the parameter $K_{x}^{y}$ (indices x, y according to the corresponding definitions), see Equations (4) and (5) when α = mel, and apply Equation (3) when the calculated parameter is the mel-opic irradiance.		

**Table 4 sensors-23-05000-t004:** Different matrix types for color sensing according to [36].

Nr.	Size	Content
1	3 × 3	[R G B]
2	3 × 5	[R G B RGB 1]
3	3 × 7	[R G B RG RB GB 1]
4	3 × 8	[R G B RG RB GB RGB 1]
5	3 × 10	[R G B RG RB GB $R^{2}$ $G^{2}$ $B^{2}$ 1]
6	3 × 11	[R G B RG RB GB $R^{2}$ $G^{2}$ $B^{2}$ RGB 1]
8	3 × 14	[R G B RG RB GB $R^{2}$ $G^{2}$ $B^{2}$ RGB $R^{3}$ $G^{3}$ $B^{3}$ 1]
9	3 × 16	[R G B RG RB GB $R^{2}$ $G^{2}$ $B^{2}$ RGB $R^{2}$G $G^{2}$B $B^{2}$R $R^{3}$ $G^{3}$ $B^{3}$]
10	3 × 17	[R G B RG RB GB $R^{2}$ $G^{2}$ $B^{2}$ RGB $R^{2}$G $G^{2}$B $B^{2}$R $R^{3}$ $G^{3}$ $B^{3}$ 1]
11	3 × 19	[R G B RG RB GB $R^{2}$ $G^{2}$ $B^{2}$ RGB $R^{2}$G $G^{2}$B $B^{2}$R $R^{2}$B $G^{2}$R $B^{2}$G $R^{3}$ $G^{3}$ $B^{3}$]
12	3 × 20	[R G B RG RB GB $R^{2}$ $G^{2}$ $B^{2}$ RGB $R^{2}$G $G^{2}$B $B^{2}$R $R^{2}$B $G^{2}$R $B^{2}$G $R^{3}$ $G^{3}$ $B^{3}$ 1]
13	3 × 22	[R G B RG RB GB $R^{2}$ $G^{2}$ $B^{2}$ RGB $R^{2}$G $G^{2}$B $B^{2}$R $R^{2}$B $G^{2}$R $B^{2}$G $R^{3}$ $G^{3}$ $B^{3}$ $R^{2}$GB R$G^{2}$B RG$B^{2}$]

**Table 5 sensors-23-05000-t005:** Properties of the lamp types for matrix optimization.

Lamp Type	Tungsten Halogen	Xenon-2	CFL 3000 K	CFL 5000 K	FL 627	FL 645	LED C3L	LED c3N	LED RGBWW4500
CCT (in K)	2762	4100	2640	4423	2785	4423	2640	4580	4500
Duv ($\;\cdot \;10^{-3}$)	3	7.2	0.61	2.2	1.8	2.2	6.3	1.3	−1.0
CIE $R_{9}$	85	−86	48	−60	−72	−60	−28	−39	36
CIE $R_{a}$	97	67	90	68	64	68	67	69	90

**Table 6 sensors-23-05000-t006:** Chromaticity differences $\Delta u^{\prime }v^{\prime }$ for different matrix types for the lamp types of the training set.

Name	Xenon-2	CFL-930	CFL-5K	FL627	FL645	C3L-pcLEDs	C3N-pcLED	RGBWW4500 K	Max
$\Delta u^{\prime }v_{3\times 3}^{\prime }\;\cdot \;10^{-3}$	7.4	7.7	1.5	4.5	1.5	0.42	2.7	8.3	8.3
$\Delta u^{\prime }v_{3\times 5}^{\prime }\;\cdot \;10^{-3}$	4.8	48	12	41	12	51	16	16	51
$\Delta u^{\prime }v_{3\times 7}^{\prime }\;\cdot \;10^{-3}$	8.5	5.3	$3.1\times 10^{-4}$	1.6	$1.8\times 10^{-3}$	7.0	20	6.4	20
$\Delta u^{\prime }v_{3\times 8}^{\prime }\;\cdot \;10^{-3}$	6.9	32	11	40	11	48	7.0	16	48
$\Delta u^{\prime }v_{3\times 10}^{\prime }\;\cdot \;10^{-3}$	2.9	4.5	$4.4\times 10^{-4}$	0.92	$8.1\times 10^{-4}$	12	$2.8\times 10^{-5}$	3.8	12
$\Delta u^{\prime }v_{3\times 11}^{\prime }\;\cdot \;10^{-3}$	7.9	$2.8\times 10^{-7}$	2.2	37	2.2	42	12	8.1	42
$\Delta u^{\prime }v_{3\times 15}^{\prime }\;\cdot \;10^{-3}$	7.1	10	$9.8\times 10^{-4}$	7.8	$3.5\times 10^{-3}$	$5.4\times 10^{-5}$	1.7	6.7	10
$\Delta u^{\prime }v_{3\times 16}^{\prime }\;\cdot \;10^{-3}$	8.4	5.7	$5.9\times 10^{-7}$	3.0	$4.1\times 10^{-3}$	2.5	1.9	10	10
$\Delta u^{\prime }v_{3\times 17}^{\prime }\;\cdot \;10^{-3}$	8.4	5.7	$5.9\times 10^{-9}$	3.0	$4.1\times 10^{-3}$	2.5	1.9	10	10
$\Delta u^{\prime }v_{3\times 19}^{\prime }\;\cdot \;10^{-3}$	7.2	9.1	$3.6\times 10^{-2}$	4.3	$3.6\times 10^{-3}$	$9.3\times 10^{-6}$	5.7	7.5	9.1
$\Delta u^{\prime }v_{3\times 20}^{\prime }\;\cdot \;10^{-3}$	7.2	9.1	$3.6\times 10^{-2}$	4.3	$3.6\times 10^{-2}$	$9.3\times 10^{-6}$	5.7	7.5	9.1
$\Delta u^{\prime }v_{3\times 22}^{\prime }\;\cdot \;10^{-3}$	6.6	8.9	$4.8\times 10^{-4}$	2.2	$1.8\times 10^{-3}$	$2.3\times 10^{-4}$	3.2	7.2	8.9

**Table 7 sensors-23-05000-t007:** Optimum matrix (3 × 3) for the transformation RGB to XYZ, as well as the formula for the calculation of the illuminance $E_{v}$ from the RGB signals.

	(19) $$E_{v}=683\times \left(1.0001\;\cdot \;E_{v,RGB}-0.0066\right)$$
$E_{v}$(lx) from R, G, B	(20) $$E_{v,RGB}=0.6802\;\cdot \;R+0.3651\;\cdot \;G+0.1751\;\cdot \;B$$
	$R^{2}=1.0$; RMSE=0.15
Matrix 3 × 3 in case of the 9 light sources of training set	(21) $$\left(\begin{array}{ccc} 5.73\times 10^{5} & -4.17\times 10^{5} & -2.27\times 10^{5}\\ 3.64\times 10^{5} & -4.50\times 10^{4} & -3.26\times 10^{5}\\ -6.61\times 10^{4} & 2.04\times 10^{5} & 1.13\times 10^{6} \end{array}\right)$$

**Table 8 sensors-23-05000-t008:** Verification of optimal matrix (3 × 3) in Table 7 with 8 *LED*s plus mixed daylight spectra.

Name	pcLED-WWG3	pcLED-WWVG1	vpcLED- WWEX4	pcLED-NWVG1	pcLED-NWEXCV2	pcLED-CWEX2	R12GB12-5000 K	TL-pcLED-30%
CCT (K)	2801	3105	2969	4614	3942	5059	5001	4391
CRI $R_{a}$	84.06	85.63	94.41	90.91	93.09	95.99	89.73	89.85
*x*	0.4434	0.4217	0.4252	0.3569	0.3764	0.3439	0.3449	0.3644
*y*	0.3929	0.3841	0.3765	0.3608	0.3550	0.3557	0.3495	0.3650
$E_{v,new}$	752.29	752.54	754.17	751.40	754.79	750.01	747.60	751.94
$x_{xew}$	0.4434	0.4217	0.4252	0.3546	0.3779	0.3368	0.3319	0.3665
$y_{new}$	0.3929	0.3841	0.3765	0.3643	0.3698	0.3571	0.3529	0.3696
$\Delta u^{\prime }v^{\prime }$	1.36 $\times \;10^{-10}$	1.20 $\times \;10^{-10}$	1.68 $\times \;10^{-10}$	3.15 $\times \;10^{-3}$	8.76 $\times \;10^{-3}$	5.32 $\times \;10^{-3}$	10.1 $\times \;10^{-3}$	2.42 $\times \;10^{-3}$
$\Delta E_{v}$ in %	0.31	0.34	0.56	0.19	0.64	0.00	-0.32	0.26

**Table 9 sensors-23-05000-t009:** Optimal matrix (3 × 3) for transforming RGB to XYZ and formula for calculating illuminance $E_{v}$ from RGB signals in case of pure daylight.

	(22) $$E_{v}=683\times \left(0.9987\times E_{v,RGB}-0.1124\right)$$
$E_{v}$(lx) from R, G, B	(23) $$E_{v,RGB}=0.6333\times R+0.4804\times G$$
	$R^{2}=1.0$; RMSE=0.04
Matrix 3 × 3 in the case of the 9 light sources of the training set	(24) $$\left(\begin{array}{ccc} 1.8558 & -1.6603 & 1.4322\\ 1.1084 & -0.23047 & 0.53993\\ 0.53135 & -0.97596 & 6.0956 \end{array}\right)$$

**Table 10 sensors-23-05000-t010:** Verification of the 3 × 3 matrix for daylight spectra (trained on a sunny day, verified on a overcast day).

Sampling Time	07:27:17	10:03:31	11:06:01	12:03:19	13:05:50	14:07:40	15:10:10	16:12:41	19:14:58
CCT in K	12,464	8033	6313	5651	5914	5853	5470	8174	16,066
$E_{v,measure}$ in lx	401	9397	23,940	58,212	39,463	43,150	71,006	15,058	461
$y_{measure}$	0.2831	0.3066	0.3293	0.3413	0.3364	0.3376	0.3449	0.3081	0.2705
$E_{v,processed}$ in lx	400.60	9403.02	23,944.07	58,205.57	39,465.20	43,149.59	70,987.13	15,070.62 4	60.41
$\Delta u^{\prime }v_{measure-calculate}^{\prime }\times 10^{-4}$	4.75	$2.81\times 10^{-2}$	1.74	1.20	1.82	1.96	0.596	4.03	8.57

**Table 11 sensors-23-05000-t011:** Measured and simulated spectra (4214 in total).

No.	Type of light sources and their parameter ranges CCT = 2201 – 17,815 K; −1.467× $10^{-2}$ < Duv < 1.529× $10^{-2}$ 0.0797 < *z* < 0.4792; 80 < CIERa < 100; 0.2784 < MDER < 1.46
1.	Conventional incandescent lamps + filtered incandescent lamps
2.	Fluorescent tubes + Compact fluorescent lamps
3.	LED lamps + LED luminaires
4.	Daylight (CIE-model + measurements)
5.	Mixtures (DL+LED) − [ mixture ratio = 10–90% ]
6.	Mixtures (DL+FL) − [ mixture ratio = 10–90% ]

**Table 12 sensors-23-05000-t012:** Parameters *a*, *b*, *c*, and *d* for the fit function in Equation (25) and the correlation coefficient $R^{2}$.

Parameter	a	b	c	d	$R^{2}$
Value	0.53806	2.3212	−0.37585	−0.85808	0.968

**Table 13 sensors-23-05000-t013:** Verification of the mDER and mEDI values with the color sensor under test on the basis of the 8 spectra in Figure 10 set to 750 lx.

Name	pcLED-WWG3	pcLED-WWVG1	pcLED-WWEX4	pcLED-NWVG1	pcLED-NWEXCV2	pcLED-CWEX2	R12GB12-5000 K	TL-pcLED-30%
CCT (K)	2801	3105	2969	4614	3942	5059	5001	4391
CIE $R_{a}$	84.06	85.63	94.41	90.91	93.09	95.99	89.73	89.85
*x*	0.4434	0.4217	0.4252	0.3569	0.3764	0.3439	0.3449	0.3644
*y*	0.3929	0.3841	0.3765	0.3608	0.3550	0.3557	0.3495	0.3650
$E_{v}$ (lx)	750	750	750	750	750	750	750	750
$mDER_{original}$	0.46	0.51	0.53	0.75	0.67	0.83	0.83	0.71
$mEDI_{original}$	347.67	383.55	395.93	562.91	500.06	625.10	625.79	532.40
$E_{v,new}$ (lx)	752.29	752.54	754.17	751.40	754.79	750.01	747.60	751.94
$x_{new}$	0.4434	0.4217	0.4252	0.3546	0.3779	0.3368	0.3319	0.3665
$y_{new}$	0.3929	0.3841	0.3765	0.3643	0.3698	0.3571	0.3529	0.3696
$mEDR_{new}$	0.46	0.53	0.54	0.74	0.66	0.81	0.83	0.69
$mEDI_{new}$	346.28	396.00	403.79	554.58	500.85	604.47	621.78	521.19
$\Delta u^{\prime }v^{\prime }$	1.36 × $10^{-10}$	1.20 × $10^{-10}$	1.68 × $10^{-10}$	3.15 × $10^{-3}$	8.76× $10^{-3}$	5.32 × $10^{-3}$	1.01 × $10^{-2}$	2.42 × $10^{-3}$
$\Delta E_{v}$ in %	0.31	0.34	0.56	0.19	0.64	0.00	−0.32	0.26
ΔmEDR in %	−0.7	2.9	1.4	−1.7	−0.5	−3.3	−0.3	−2.4
ΔmEDI in %	0.40	3.25	1.99	1.48	0.16	3.30	0.64	2.10

## Data Availability

All data generated or analyzed to support the findings of the present study are included this article. The raw data can be obtained from the authors upon reasonable request.

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
