# Peer review of "Determination and Measurement of Melanopic Equivalent Daylight (D65) Illuminance (mEDI) in the Context of Smart and Integrative Lighting"

_sensors, 2023, doi:10.3390/s23115000_

Round 1
Reviewer 1 Report
In this work, the authors have investigated the calculation method of non-visual parameters via RGB sensors and the feasibility is demonstrated methodically. The “non-visual light effects” is introduced first and then definition of the non-visual input parameters is provided. The next section of the paper involves performing a prediction using a straightforward computational model of non-visual quantities mEDI 179 and mDER. Additionally, a feasibility test of an RGB sensor is conducted based on this model in order to verify its accuracy for lighting applications. This work is of interest to other researchers in scientific and engineering community of smart and integrative lighting. However, there are a few comments to be addressed. The detailed comments are as follows:
1) In abstract and introduction, the authors write: “In the context of intelligent and integrative lighting, in addition to the need for color quality and brightness, the non-visual effect is essential.”…“Control or regulation of modern semiconductor based lights (LED - OLED) and window systems (daylight systems) with the help of sensors in the room (e.g. presence sensors, position sensors, light and color sensors)”. The general reference list seems a bit thin, considering the evolution in the field within the recent years. To give the readers a much broader view, recent developments related to LEDs lighting, such as Laser & Photonics Reviews 2023, 17, 2200455 (https://doi.org/10.1002/lpor.202200455); Optics Express 27(12), A669 (2019); etc. should be added, so that the readers can be clear about the state-of-the-art of this topic.
2) What does the physical quantity in the formula represent in Section 2? Please provide a detailed derivation process of the formula
3) From Figure 6, it can be observed that the three types of sensitivity in RGB sensors are different, so it is necessary to illustrate the effect of sensitivity on the calculation of non-visual effects parameters
4) Could the authors explain the effect of matrix transformation on the error of signal processing?
5) What has caused the difference between chromaticity (x, y, z) of the RGB sensor and that derived from the CIE calculation?
6) In this manuscript, the text is interrupted by figures and tables. It is recommended to modify accordingly to improve the readability.
Author Response
Dear Reviewer,
We would like to thank you for taking your valuable time to read and improve our manuscript!
Your comments, questions and recommendations are very useful for us to revise this manuscript.
We have been receptive to your ideas to build a better version. And all our answers are written in blue text in the attached file directly under your questions.
Thanks again and best regards,
Authors of the article

Reviewer 2 Report
Dear Authors,
Thank you for preparing the manuscript. I found inside a good piece of work. As far as I understood it was focused on calculations with some data gathered for the first step of the research. The content seems clear for me, but I found some flaws that should be improved or clarified. For this reason I recommend major revisions.
In my opinion the weakest part of this manuscript is the literature review. In my opinion it is to wide and chaotic, but on the contrary it does not contain any references to works devoted to testing the reliability this type of sensors (even in other applications).
Below you can find my main coments:
1. In the abstract you mention "appropriate matrixing process" and "proper illumination" - these adjectives are controversial to many researchers. Specific criteria should be refferd. Also dofferent construction of the sentece could be a solution.
2. In the introduction you collected some data to build the table 1. I am really surprised with the content. According to the standards lighting criteria are much wider (you do not mention uniformity of illuminance, glare, illuminances on the walls etc.) and what is more - you stated that illuminance should be above 850lx based on selected references, but not EN12464-1. In the European standard for interior lighting there is a chain of possible illuminances starting much below 850 lx. You reffer to this standard only in case of CRI. It is really surprising. I am strongly against this table - it is misleading and incorrect in this version.
2. You do not mention to WELL building standard, which brings clear requirements for melanopic illuminance. It is surprising, because - according to my knowledge - this standard is the only one in the world, which gives some procvedures and required limits for integrative lighting.
3. Page 2 line 46 - word "newline" appeared in the beginning of the line
4. the first research question on page 2 - in my opinion you should make a deeper literature review to answer it or erase that question by making assumption that you focus on the specific parameters as mentioned in the title of the manuscript. There are more parameters proposed and used by the researchers in the research considering non-visual effects (for example the one in WELL Building Standard, but also those promoted by Lighting Resrearch Center based on circadian stimulus). You haven't analyzed them at all - you have only focused on the CIE 026:2018. In my opinion there is nothing wrong in using the parameters from this CIE publication, but the literature review should present also other approaches, if you ask such question.
5. page 4 lines 132 -133 - you mention the "individual needs of light users in dependence on weather conditions, individual human characteristics, time and context" - what are these needs? Can you give numbers and schemes justified by research? Are there any agreed and supported by deep research? Inb my opinion this sentence needs revision or justification by literature.
6. on page 6 line 178 you wrote mEDR instead of mDER
7. on page 11 Table 5 - how can illuminance be treated as the parameter of the lamp? It is incorrect.
8. In paragraph 3.3 - as I understand - you performed some experiments and derived 2 versions of the matrix. Which one was used in the final calculations of the mixed light spectra?
9. Table 8 may not clear for the reader. Please, explain in more details.
10. Page 14 - I have doubts for the "rainy day" (line 316 and caption of the table) - it is not typical terminology in lighting. According to CIE terminology we have clear sky, overcast and intermediate sky. COnsidering the values of illuminancereaching over 70klx it was probably not overcast. What were the weather conditions according to terminology used by researchers working with daylighting? You may also provide weather conditions data from freely available tools.
11. In line 337 you state, that "matrices were optimized and found" - I suggest to erase "optimized"
12. Please, clairify the titlem, abstract and procedure of the research, for example by preparing the flowchart of the research. From your manuscript it could be understood that you have performed the measurements with the integrative lighting, According to what I understood you made some experiments with daylight and electric light sources, but later the research consists only calculations.
In my opinion the language does not require improvements.
Author Response

(The authors gave the same response as above.)

Reviewer 3 Report
The present work makes a theoretical study of the determination and measurement of daylight equivalent melanopic illuminance in the context of smart and integrative lighting. Overall the work is well-written and designed. However, some modifications are required before publication. The first point that needs significant improvement is the Introduction. In addition to being extensive, the Introduction is somewhat confusing, and the novelty of the work is not apparent. I suggest the authors define the points the manuscript addresses more clearly. Tables 8 and 13 are too small and need to be improved. Finally, the Conclusion and Discussion section neither concludes nor discusses the work, so I suggest it be improved before publication.
The English used in this work is consistent and well-written, however, there is a need for a final revision to improve and correct small flaws.
Author Response

(The authors gave the same response as above.)

Round 2
Reviewer 2 Report
Dear Authors,
Thank you for your response and clarifications. Some of them seem to be enough, however I am still confused with the table 1. With this content people who are not aware of this standard could think there are no technical standards on that issue. On the contrary researchers who are aware of them woukld react as I did. There is nothing wrong in limiting the analysis to specific conditions, but the background should not leave any doubts. Considering the table 1. I udnerstand your point of view, however standards are a good practice and they also come from research. You state "Table 1 was inconsistent because we mixed the latest research results with the EN 12464 value. In our original intention we would like to mention some rather new research literature and not the lighting standards (e.g., EN 12464 and others). To this article focusing on the measurement methods for determining mDER and mEDI, we would like to mention only information from recent literature in Table 1 and added some literature values for luminance on ceiling and walls, hopefully in your sense (see your criticism above)."
It is difficult to say that reffered publications are the "latest research" if the standard comes from 2021 and publications start from 1998. I believe it would be much easier to state, that based on technical requirements you select only some types of interiors, where specific technical requirements are valid and that's all (so make the assumption that you select some type of interior).
As you erased illuminance from table 5, I suggest to do the same in Table 8.
Author Response
Dear Reviewer 2,
Thank you for taking your valuable time to read and improve our manuscripts!
You're right. As a result, we will add the following sentences in the revision in lines 35 to 41 of Page 2 and delete the sentence above Table 1 of the old version:
"As a result of this research process, international and national standards (e.g. EN 12464 [10]) have been formulated for lighting design and development of interior lighting systems. For special applications (e.g., exhibitions, meeting rooms in prestigious buildings, treatment rooms in medical centers and hospitals), the following values in Table 1 [11-18] from research studies would be used".
As well, the values of illuminance Ev in lx have also been removed from Table 8 (similar to Table 5 in Round 1). Thanks again for this recommendation!
The authors of the article
